applied mathematics/psychology/behaviour

Prisoner's Dilemma, evolutionary game theory, psychopathy, group differences

**Author for correspondence:**
Martina Testori
e-mail: m.testori@soton.ac.uk

# How group composition affects cooperation in fixed networks: can psychopathic traits influence group dynamics?

Martina Testori[1], Rebecca B. Hoyle[1]
and Hedwig Eisenbarth[2,3]

[1]School of Mathematical Sciences, and [2]School of Psychology, University of Southampton, Southampton, UK
[3]School of Psychology, Victoria University of Wellington, Wellington, New Zealand

MT, 0000-0001-7292-7129; RBH, 0000-0002-1645-1071;
HE, 0000-0002-0521-2630

Static networks have been shown to foster cooperation for specific cost–benefit ratios and numbers of connections across a series of interactions. At the same time, psychopathic traits have been discovered to predict defective behaviours in game theory scenarios. This experiment combines these two aspects to investigate how group cooperation can emerge when changing group compositions based on psychopathic traits. We implemented a modified version of the Prisoner's Dilemma game which has been demonstrated theoretically and empirically to sustain a constant level of cooperation over rounds. A sample of 190 undergraduate students played in small groups where the percentage of psychopathic traits in each group was manipulated. Groups entirely composed of low psychopathic individuals were compared with communities with 50% high and 50% low psychopathic players, to observe the behavioural differences at the group level. Results showed a significant divergence of the mean cooperation of the two conditions, regardless of the small range of participants' psychopathy scores. Groups with a large density of high psychopathic subjects cooperated significantly less than groups entirely composed of low psychopathic players, confirming our hypothesis that psychopathic traits affect not only individuals' decisions but also the group behaviour. This experiment highlights how differences in group composition with respect to psychopathic traits can have a significant impact on group dynamics, and it emphasizes the importance of individual characteristics when investigating group behaviours.

# 1. Introduction

Human interactions are characterized by complex networks of individuals and relationships. Cooperation is one of the basic interactions among them, and for decades researchers have tried to explain how it evolves and which circumstances can boost cooperative behaviours. Evolutionary game theory offers numerous examples of how to foster cooperation by modelling human actions in various situations. Mechanisms such as the evolution of communities in networks, reputation systems, both altruistic and institutional punishments, iteration of games over time have all been proven to sustain cooperation, both theoretically and experimentally [1–6]. A recent experiment [7] confirmed an important theoretical argument claiming that: 'natural selection favours cooperation, if the benefit of the altruistic act, $b$, divided by the cost, $c$, exceeds the average number of neighbours, $k$, which means $b/c > k$' [8, p. 502]. In their experiment, Rand and co-workers [7] proved that, satisfying the benefit–cost condition and adopting a static instead of a well-mixed network, cooperation was not only fostered but also maintained over time. All the above-mentioned works address the study of cooperation by looking at how external factors influence and promote collaborative behaviours. In the current study, however, we are interested in exploring how individual characteristics interact with these exogenous mechanisms, when considering group dynamics.

Human personality traits and group dynamics have been analysed in the study of team work and effectiveness, especially in the workplace. Existing research has clearly established that group personality composition affects group performance [9–12]. Teams with higher extraversion and emotional stability were found to enhance productivity and team viability [10]. Conscientiousness, agreeableness and openness to experience at the group level were positively correlated with team performance, as were the differences in extraversion and emotional stability among group members [11]. In small groups, extraversion, both at the individual and at the group level, predicted task focus and group performance [9].

Thus, numerous analyses have been implemented to disentangle possible connections among cooperation, group performance and personality traits. However, no previous contribution has looked at the relationship between psychopathic traits and group dynamics, which is the focus of our study.

In his work, Cleckley [13] described the construct of psychopathy as characterized by a constellation of personality traits including superficial charm, lack of remorse, guilt and fear, poor impulse control, emotional detachment and impairment in building solid relationships, as well as high levels of manipulativeness, dishonesty, low empathy and callousness. Several studies have examined the effect of psychopathy on cooperation, especially using game theory. One of the first studies using game theory to investigate psychopathic traits adopted the Prisoner's Dilemma game, and surprisingly it found that males high in primary and secondary psychopathy were not more likely to defect than those low in such traits [14]. Nevertheless, later studies reported significant negative correlation between psychopathic traits and cooperation [15], and male prison samples showed a decreased cooperativeness among those high in psychopathy [16]. High levels of impulsivity were also found to be strongly predictive of defective behaviours in the general population [17]. Although prior research has looked at the relationship between psychopathic traits and cooperation, no study has yet examined how psychopathy affects group cooperation.

To address this gap, we conducted a laboratory experiment using the experimental design proposed by Rand *et al.* [7], including participants' psychopathic traits. We were interested in how the introduction of high psychopathic individuals would affect cooperation at the group level, in an environment that has been proven to foster and maintain cooperation. Participants' psychopathic traits were assessed before the laboratory experiment, and groups were formed in such a way as to have either 0%, 20% or 50% of high psychopathic people in each group. Our goal was to find an answer to the research question: 'Do higher psychopathic people in the general population affect group dynamics?'.

While several studies have looked at the general population when investigating psychopathy [15,17], in this experiment, we used a subsample of the general population, composed of undergraduate students, in which the variation of psychopathic traits was quite small. In this way, we observed whether even small changes in psychopathic traits can have an impact not only on individuals' strategy but also on group dynamics. Based on previous findings and on the depiction of psychopathic traits, our hypothesis is that groups with a greater density of high psychopathic individuals will show less cooperative behaviours, when compared with groups with low or zero density of high psychopathic participants.

# 2. Methods

## 2.1. Sample

Participants were recruited among Southampton University students. A total of 305 participants filled in the online PPI-R-40 questionnaire [18]. Among them, 201 participants took part in the laboratory experiment after being invited by the researcher. This selection was due to their availability to take part in the laboratory experiment and on their consistency in filling in the online questionnaire. The first two sessions were pilot versions (used to check the functionality of the experiment) and were not included in the final sample, which was composed of 190 participants (115 female, age: $M = 23.31$, s.d. $= 4.68$). Each participant took part to the experiment only once. Participants gave informed consent for participating in a laboratory game and ethical approval was obtained from the Faculty of Social, Human and Mathematical Sciences at the University of Southampton.

## 2.2. Personality measures

The PPI-R is a 154-item self-report questionnaire [19] on psychopathic traits with eight sub-scales: Machiavellian egocentricity, social potency, coldheartedness, carefree non-planfulness, fearlessness, blame externalization, impulsive non-conformity and stress immunity. Seven of the eight sub-scales can be grouped into two main factors: fearless dominance and self-centred impulsivity, while coldheartedness is considered as an additional factor. Eisenbarth *et al.* [18] proposed a 40-item version of the PPI-R, which was used in this study. A recently developed method, the IRS-10, allowed us to test the response reliability of participants in the PPI-R-40 [20]. Participants with IRS-10 scores above the cut-off (99th percentile) were deemed to have completed the PPI-R-40 in an inconsistent and therefore unreliable manner and were eliminated from analysis, leading to the exclusion of two participants.

To compare the psychopathic traits of our sample with data from previous studies ([21] (study 1) and [22] (study 2)), we calculated Cohen's [23] approximate metrics for group differences, where $d = 0.2$ is considered a weak difference, $d = 0.5$ is medium, and $d = 0.8$ or higher is large. We compared the cumulative measure of psychopathy PPI-R-SUM, and the three subcategories of fearless dominance, self-centred impulsivity and coldheartedness (see table 1 for results). Our sample reports smaller values compared with the two reference studies considered (thus the negative sign of the Cohen's $d$) and such differences are all large ($d > 0.8$).

## 2.3. Experimental design

In this experiment, we used the design implemented by Rand *et al.* [7]. Participants were arranged on a ring connected to one neighbour on each side, for a total of $k = 2$ links per player. They had an initial endowment of 100 points and they played a repeated cooperation game over 50 rounds. In each round, they had to choose whether to defect, by doing nothing or to cooperate, by paying a cost of $c = 10$ points per neighbour to give each of them a benefit of $b = 60$ points ($b/c = 6 > k = 2$). This setting was chosen according to the Rand *et al.* [7] findings, where this ratio showed a more constant maintenance of cooperation over rounds. Each player made a single decision in each round, meaning that they could not cooperate with one neighbour and defect with the other. At the end of each round, participants were shown their neighbours' decisions, as well as the cumulative and the round pay-off earned by themselves and by each neighbour. Participants were assigned to a position on the network and they did not change neighbours throughout the entire game. The number of rounds was not shown during the game in order to simulate an infinite game and to avoid an end-of-game effect, although they were initially informed of the duration of the game (roughly 40 min) and the total number of rounds.

## 2.4. Experimental manipulation

High psychopathic individuals were defined as those participants scoring in the top quartile of the PPI-R-40 total score for our sample (PPI-R-40 total score $>101$), while all other players were considered low psychopathic. The percentage of highly psychopathic individuals per session was manipulated in order to obtain three conditions: *high*, *low* and *zero* density (table 2). High and low psychopathic

**Table 1.** Descriptive statistics for the participants sample. Gender: 1 = male, 2 = female; nationality: 1 = UK, 2 = other; maximize.yourself, maximize.links, behaviour.neighbours: 0 = no, 1 = yes.

|  | min | max | mean | standard deviation | Cohen's $d$ study 1 | Cohen's $d$ study 2 |
|---|---|---|---|---|---|---|
| age | 19 | 52 | 23.32 | 4.66 | — | — |
| gender | 1 | 2 | 1.60 | 0.49 | — | — |
| nationality | 1 | 2 | 1.40 | 0.49 | — | — |
| cumulative psychopathy measure | 85 | 111 | 98.15 | 5.05 | −2.11 | −2.28 |
| fearless dominance | 29 | 46 | 36.66 | 2.55 | −2.19 | −2.90 |
| self-centred impulsivity | 27 | 45 | 37.16 | 3.20 | −2.83 | −2.34 |
| coldheartedness | 7 | 16 | 11.82 | 1.98 | −1.43 | −1.11 |
| maximize.yourself | 0 | 1 | 0.75 | 0.43 | — | — |
| maximize.links | 0 | 1 | 0.59 | 0.49 | — | — |
| behaviour.neighbours | 0 | 1 | 0.86 | 0.35 | — | — |

**Table 2.** Descriptive statistics for the three conditions.

| conditions | % of high psychopathic individuals | sessions | participants | participants per session |
|---|---|---|---|---|
| *high* | M = 50.75%, s.d. = 5.75 | 8 | 73 | median = 9, range = {8, 11} |
| *low* | M = 20%, s.d. = 9.82 | 6 | 55 | median = 9, range = {7, 11} |
| *zero (baseline)* | M = 0%, s.d. = 0 | 9 | 62 | median = 7, range = {5, 9} |

participants were arranged on the ring in such a way as to avoid clusters of high or low psychopathic players, i.e. high psychopathic individuals were evenly distributed around the ring in each session. The difference in groups' size did not affect the cooperation evolution over the 50 rounds in any of the three conditions (Pearson's correlation $p$-value = {0.67, 0.50, 0.29}, respectively, for the three conditions).

## 2.5. Experimental procedure

First, participants filled in an online questionnaire to assess their psychopathic traits and gave their consent to be contacted for a laboratory experiment. Participants were told the questionnaire was a personality test and no specific instructions were released regarding the effect of the questionnaire on the invitation to the laboratory experiment. Participants were then invited by the researcher to attend a laboratory session, according to their personality score. Each participant was randomly assigned to a computer station according to their psychopathic scores, and they were not able to see each others' screens. Participants received a £10 fixed rate for completing the experiment, plus an additional £1 for every 1000 points earned during the game ($M = 2.75$, s.d. = 1.13). Players read the instructions on the screen and they then played one practice round, which was not included in the final pay-off. After having completed the game, they filled in a short questionnaire to assess their understanding of the game and to describe their strategy and predispositions during the game. Three main questions were asked during this follow-up questionnaire: 'Did you try to achieve the highest score for yourself?', (variable: maximize.yourself), 'Did you try to obtain the highest score for yourself AND your links?', (variable: maximize.links) and 'Did you adjust your strategy according to your neighbours' previous actions?', (variable: behaviour.neighbours). Participants' answers were then used in the analysis to observe which motivations were more influential in the strategies adopted.

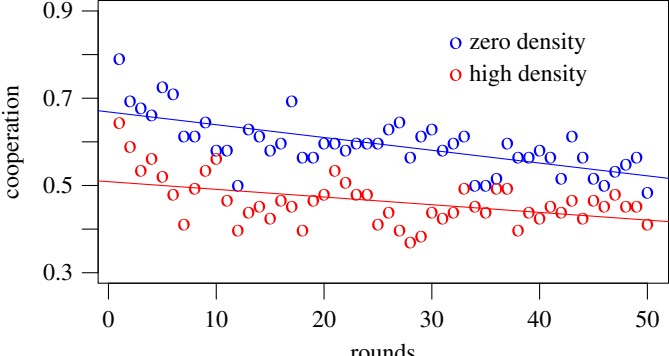

**Figure 1.** Cooperation variable calculated as the average cooperation per group per round. Zero density condition promotes cooperation compared with high density groups. The fraction of subjects cooperating in each round is shown averaged over groups, for the zero (blue circles) and high (red circles) density conditions.

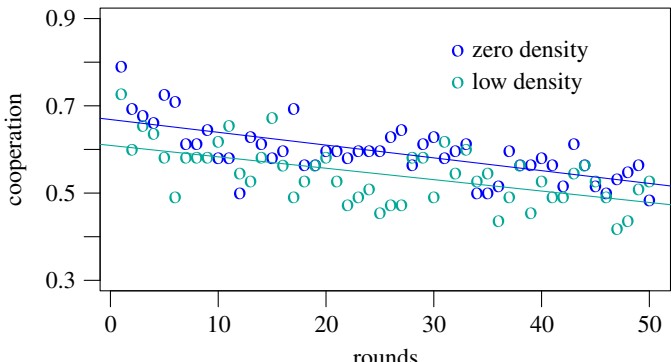

**Figure 2.** Cooperation variable calculated as the average cooperation per group per round. Cooperation in the zero and low density conditions is not significantly different, but we can observe an overall lower level of cooperation in the low density groups, compared with the zero density one. The fraction of subjects cooperating in each round is shown averaged over groups, for the zero (blue circles) and low (cyan circles) density conditions.

## 3. Results

Since the aim of the experiment was to investigate group variations in the three conditions, the analysis adopted the average of cooperative decisions per group (0 = defect, 1 = cooperate). Figure 1 illustrates the groups cooperation throughout the 50 rounds for the high and zero density conditions. It is evident, by looking at the overall level of cooperation, that groups composed by 50% of high psychopathic people cooperated significantly less than groups entirely composed by low psychopathic people (significance level in table 4). This result corroborated our theoretical prediction, proving the influence of psychopathic traits not only on individuals' decisions, but also on group behaviours.

On the other hand, figure 2 compares the evolution of cooperation throughout the game between the zero and the low density conditions. Although the overall cooperation between the two conditions is not largely different, the trend confirms what is reported in figure 1: groups having some high psychopathic players exhibited a consistently lower level of cooperation compared with groups entirely composed of low psychopathic individuals. Hence, both figures 1 and 2 corroborate our initial hypothesis that having high psychopathic players alters the group dynamics toward less cooperative behaviours.

To observe whether this behaviour is actually caused by the presence of high psychopathic individuals in the group, we looked at the correlation between psychopathic traits and cooperation. As table 3 reports, having higher scores in the fearless dominance sub-scale of psychopathy is correlated with less cooperative behaviours. This supports the claim that high psychopathic individuals show less cooperative behaviour compared with low psychopathic individuals. In particular, our results suggest that the fearless dominance component of psychopathy is the driving factor of such divergence in behaviours.

**Table 3.** Bivariate correlation matrix among cooperation, personality traits and social demographics. Gender: 1 = male, 2 = female; nationality: 1 = UK, 2 = other.

| | fearless dominance | self-centred impulsivity | coldheartedness | age | gender | nationality | maximize links | maximize yourself | behaviour neighbours |
|---|---|---|---|---|---|---|---|---|---|
| fearless dominance | 1 | — | — | — | — | — | — | — | — |
| self-centred impulsivity | 0.01 | 1 | — | — | — | — | — | — | — |
| coldheartedness | 0.10 | −0.08 | 1 | — | — | — | — | — | — |
| age | 0.04 | −0.04 | −0.09 | 1 | — | — | — | — | — |
| gender | 0.13* | 0.19** | 0.10 | 0.04 | 1 | — | — | — | — |
| nationality | 0.15* | 0.01 | 0.05 | −0.26*** | 0.07 | 1 | — | — | — |
| maximize links | −0.02 | 0.17 | −0.06 | −0.09 | −0.01 | −0.03 | 1 | — | — |
| maximize yourself | 0.03 | 0.00 | −0.01 | 0.20** | 0.05 | 0.03 | −0.33*** | 1 | — |
| behaviour neighbours | −0.05 | 0.04 | −0.05 | 0.12 | −0.05 | −0.01 | −0.03 | 0.25*** | 1 |
| cooperation | −0.16* | 0.06 | 0.05 | −0.11 | −0.11 | −0.17* | 0.37*** | −0.24*** | −0.31*** |

significance level: *** $<$0.001, ** $<$0.01, * $<$0.05, # $<$0.1

**Table 4.** Cooperation as a function of descriptive characteristics and motivations. Logistic linear mixed models, with random effect at the subjects level.

| | model 1 | model 2 | model 3 | model 4 | model 5 |
|---|---|---|---|---|---|
| age | — | 0.04 | 0.04 | 0.04 | 0.03 |
| | — | (0.03) | (0.03) | (0.03) | (0.03) |
| gender | — | −0.32 | −0.32 | −0.32 | −0.38 |
| | — | (0.27) | (0.27) | (0.27) | (0.28) |
| nationality | — | −0.77** | −0.68* | −0.68* | −0.60* |
| | — | (0.29) | (0.29) | (0.29) | (0.29) |
| maximize.links | — | 1.33*** | 1.34*** | 1.34*** | 1.30*** |
| | — | (0.29) | (0.28) | (0.28) | (0.28) |
| maximize.yourself | — | −0.20 | −0.23 | −0.23 | −0.24 |
| | — | (0.34) | (0.34) | (0.34) | (0.34) |
| behaviour.neighbours | — | −2.40*** | −2.35*** | −2.36*** | −2.37*** |
| | — | (0.43) | (0.43) | (0.43) | (0.43) |
| condition *low* | −0.37 | — | −0.42 | −0.45 | −0.40 |
| | (0.40) | — | (0.34) | (0.36) | (0.36) |
| condition *high* | −0.92* | — | −0.80* | −1.01** | −1.06** |
| | (0.37) | — | (0.32) | (0.33) | (0.35) |
| fearless dominance | — | — | — | — | −0.08 |
| | — | — | — | — | (0.05) |
| self-centred impulsivity | — | — | — | — | 0.06 |
| | — | — | — | — | (0.04) |
| coldheartedness | — | — | — | — | 0.09 |
| | — | — | — | — | (0.07) |
| round | — | — | −0.01*** | −0.01*** | −0.02*** |
| | — | — | (0.00) | (0.00) | (0.00) |
| condition *low*\* round | — | — | — | 0.00 | 0.00 |
| | — | — | — | (0.00) | (0.00) |
| condition *high*\* round | — | — | — | 0.01* | 0.01* |
| | — | — | — | (0.00) | (0.00) |
| intercept | 0.86** | 1.53 | 2.21*** | 2.31* | 2.26 |
| | (0.27) | (0.94) | (0.95) | (0.96) | (2.79) |

Standard errors in parenthesis. Significance level: ***<0.001, **<0.01, *<0.05, #<0.1.

This difference in results between conditions is confirmed when analysing the data in more detail (using logistic linear mixed models, with random effects at the subjects level). We included demographic characteristics (*age, gender* and *nationality*) to adjust for possible disparities across condition samples, and possible motivation variables (*maximize.links, maximize.yourself* and *behaviour.neighbours*) to have a more detailed representation of the dynamics. Demographics statistics of the sample are reported in table 1. The effect of the high condition (reference category: zero condition) is statistically significant (table 4, model 1), even without adjusting for possible divergences in the conditions' samples. This means that groups composed of 50% of high psychopathic subjects cooperated significantly less than groups composed only of low psychopathic people. Including the interaction terms between the conditions and the rounds, we observe an increase in the significance level of the high density condition. The interaction terms disentangle the overall effect of the conditions on cooperation from the evolution of cooperation over rounds (adjusting for the slope of cooperation). Thereby, we can see that the overall level of cooperation in the high condition is

**Table 5.** Cooperation as a function of rounds and conditions. Logistic linear mixed models with random effect for subjects. Standard errors in parenthesis.

| | (1) round 1–12 | (2) round 13–25 | (3) round 26–37 | (4) round 37–50 |
|---|---|---|---|---|
| condition *low* | −0.76 | 0.31 | −3.46* | −1.17 |
| | (0.49) | (0.81) | (1.36) | (1.49) |
| condition *high* | −1.12* | −1.77* | −4.57*** | −2.27# |
| | (0.46) | (0.74) | (1.22) | (1.35) |
| round | −0.13*** | −0.02 | −0.07** | −0.04 |
| | (0.03) | (0.03) | (0.03) | (0.02) |
| condition *low*\* round | 0.08# | −0.03 | 0.10* | 0.02 |
| | (0.04) | (0.04) | (0.04) | (0.03) |
| condition *high*\* round | 0.05 | 0.05 | 0.12** | 0.04 |
| | (0.04) | (0.04) | (0.04) | (0.03) |
| intercept | 1.81*** | 0.98# | 2.96** | 1.94 |
| | (0.34) | (0.55) | (0.94) | (1.01) |

Significance level: ***$<$0.001, **$<$0.01, *$<$0.05, #$<$0.1.

significantly lower than the one in the zero condition, but the level of cooperation is maintained more constant over time in the high density condition, compared with the zero density case. Hence, the positive sign of the interaction coefficient explains this difference in the maintenance of cooperation over time. This difference in the evolution of cooperation was evident from figure 1, and it is further analysed later in the results.

Furthermore, it is interesting to observe some other significant correlations relating participants' motivations to their strategies (table 4, model 3). Trying to maximize both the neighbours' pay-off and their own (*maximize.links*) led participants to cooperate significantly more, compared with players who did not try to achieve the best for both themselves and their links. By contrast, players who tried to maximize only their own profit (*maximize.yourself*) had a tendency to cooperate less than others, although not to a significant extent. An interesting finding arose from the *behaviour.neighbours* variable: when participants reported being influenced by their links' actions, they cooperated significantly less compared with players who were not influenced by their neighbours' decisions. Notice that neither of those variables are correlated with the inidividuals' personality measures (table 3). Finally, the data show a significantly higher cooperation of UK citizens, compared with others (nationality: 1 = UK, 2 = others).

Moreover, by including the individual personality measures in the regression model (table 4, model 5), we noticed that the individual differences do not have a statistically significant effect on cooperation. In other words, despite the individual psychopathic measures, players adopted more defective behaviours when they were part of a group half composed of high psychopathic people (i.e. high density condition). Such a result corroborates our initial hypothesis: when composed by both high and low psychopathic members, groups cooperate less regardless of members' personal level of psychopathy.

The focus of this experiment was to observe the main effect on group dynamics, when changing the group composition. As visible in figure 1 and from the results in table 4, psychopathic traits not only influenced individuals' behaviour, but they also had a strong impact on the groups' cooperation, as initially hypothesized. In order to have a better understanding of how groups acted throughout the 50 rounds, we divided the game into four consecutive subsets and observed how the high and low density groups behaved (table 5), compared with the zero density (logistic linear mixed model, random effect at subjects level).

As remarked above, the level of cooperation in the high density condition was overall significantly lower than in the zero density condition, and this trend was consistent over all 50 rounds. By contrast, the behavioural pattern in the low condition did not diverge significantly from the zero density condition, except from the third quarter of the game. In rounds 26–37 both high and low density groups cooperated significantly less than the zero density groups. However, looking at the interaction term between

**Table 6.** Pearson's correlation coefficients between rounds and overall cooperation in the three conditions. Cooperation varies only in the zero density condition.

| | round 26–50 | round 34–50 | round 38–50 |
|---|---|---|---|
| condition zero | −3.50** | −0.25 | −1.92# |
| condition low | −1.53 | −0.61 | −0.87 |
| condition high | 1.71 | −0.43 | 1.20 |

Significance level: ***<0.001, **<0.01, *<0.05, #<0.1.

conditions and rounds, we can see that it is positive for both conditions. Since the interaction term describes the differences in the slopes of the two cooperation lines (high and low conditions compared with the zero density one), the positivity of the coefficient indicates a significantly less steep decrease in cooperation in both high and low groups compared with the zero density condition in rounds 26–37.

Lastly, since the experiment adopted the experimental design proposed by Rand *et al.* [7], we were interested in observing whether our results could replicate their findings. In table 6, we considered the last half (rounds 26–50), the last third (34–50) and last quarter (38–50) of the game as analysed in Rand *et al.* [7]. Our results showed no correlation between the level of cooperation and the rounds for both low and high conditions. On the other hand, it seems that participants in the zero density condition suffered the end-of-game effect [24,25]: players show a significant decrease in cooperation towards the end of the game.

## 4. Discussion

We investigated the effect of different group composition on cooperative behaviours, looking at different density of psychopathic traits within the group members. Manipulating the group configuration, we looked at how groups with a low/high density of highly psychopathic people (20/50%) behaved, compared with groups with no highly psychopathic players (0%). We adopted the experimental design developed by Rand *et al.* [7], setting the ratio between cost and benefit of cooperation greater than the number of links each participant had ($b/c = 6 > k = 2$). We implemented this design to analyse how the introduction of high psychopathic people would affect the group behaviour, in an environment that has been shown to maintain cooperation over rounds.

Our results show that *people with higher levels of psychopathic traits do affect group dynamics*. We found a significant divergence of cooperation in those groups having a high density of high psychopathic participants compared with the zero density groups. Our findings were also robust when controlling for individuals' personality measures: belonging to a group composed of both high and low psychopathic individuals led players towards more defective strategy, regardless of their personal level of psychopathy. This has relevant implications for group settings, e.g. team work in companies or educational environments. On an individual level, psychopathy has been found to be related to counterproductive work behaviour [26] and negative impact on employees [27]. Our results therefore align with negative effects of psychopathic personality traits on individuals in the work context, but extend those findings to less cooperative behaviour in team settings. This could have implications for building and managing teams, especially when cooperative behaviour is crucial for successful team work. This result is additionally striking, considering the sample of the experiment: in contrast to previous studies on psychopathy [16], we considered psychopathic traits in a subset of the general population (undergraduate students), rather than in criminal psychopaths. Furthermore, as our sample was composed of university students, the range of psychopathy measures was very restricted, even compared with the general population [21,22]. Nevertheless, the effect of the high density condition is evident and strongly significant.

This study also highlighted that a substantial proportion of individuals high on psychopathic traits scores is necessary to affect group behaviour. Having only a small proportion of participants showing high psychopathic traits (20%) was not enough to provide a significant impact on cooperation. On the other hand, when half of the group was composed of high psychopathic participants, the group's behaviour changed significantly, showing more defections compared with groups with no high psychopathic individuals.

Another interesting aspect is the dissimilarity between the results reported by Rand *et al*. [7] and ours. Since our analysis showed no correlation between the level of cooperation and the rounds for both low and high conditions, we can state that the cooperation was maintained constant over time in these two cases. In other words, two of the three conditions of our experiment replicated Rand *et al*. [7] results: when specific network and pay-off conditions are satisfied (static network and $b/c > k$), cooperation does not fluctuate over time. Nonetheless, the zero density condition did not corroborate Rand *et al*. [7] findings, showing an end-of-game effect. However, it is hard to give an interpretation to why these differences emerged. A possible explanation could be that in the two conditions with high psychopathic traits (low and high density), the level of cooperation was already very low. Hence, it would be difficult to record an additional decrease in cooperative actions. Alternatively, the difference in the results could be explained by the different sample sizes adopted in the two experiments. While our groups were formed by maximum 11 players, Rand *et al*. [7] created much larger groups (average of 24 players per group) for the static network setting (while smaller groups—average 8—for the well-mixed network). Nevertheless, it would be interesting to address this point in future research to disentangle the end-of-game effect from other possible mechanisms not yet identified.

Furthermore, the experiment showed how some players' predispositions are important in the decision-making process. Trying to maximize both their own personal and their partners' pay-offs led people to cooperate significantly more, while individuals focused only on their personal gain were more prone to defect, although not to a significant extent. Moreover, when influenced by partners' previous actions, participants cooperated less than average. This could suggest that only partners' defective behaviours had an influence on players' decisions, driving them towards less cooperative behaviours.

Although having a small range of psychopathic traits resulted in a strong impact of such traits on group dynamics, it would be interesting to collect a larger sample of participants to have a deeper understanding on how variations of psychopathic traits influence cooperation at the group level: a larger spectrum of psychopathic traits would allow us to understand the internal dynamics of the group, investigating how cooperation evolves over rounds for high and low psychopathic players.

This study addresses an important gap in the literature regarding the effect of individuals' personality traits in a group context. Our work is one of the first experimental investigations of the effect of individual psychopathic traits on cooperation in groups, and we showed that individuals' psychopathic traits do influence group behaviours, even when only small variations are present between group participants. With this study, we aimed to integrate the effect of individual personality traits into the large body of literature investigating how to promote cooperation, to highlight how individual differences are determinant for a more comprehensive study of the evolution of cooperation.

Ethics. Ethical approval was obtained from the Faculty of Social, Human and Mathematical Sciences at the University of Southampton (ERGO no. 31209). All experimental and survey procedures followed ethical guidelines from the Declaration of Helsinki as well as guidelines of the institutional review board. All participants provided informed consent.

Data accessibility. Our data are deposited at the Dryad Digital Repository: https://doi.org/10.5061/dryad.ms57853 [28].

Authors' contributions. M.T. contributed to study conceptualization, experimental design, data collection and data preparation. All authors contributed to data analysis and report writing and gave final approval for publication.

Competing interests. The authors declare no competing interests.

Funding. We gratefully acknowledge the support of the School of Mathematical Sciences, the School of Psychology and the Institute for Life Sciences at the University of Southampton.

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
