## [Reviewer comments · Royal Society Open Science]

Review History

RSOS-181329.R0 (Original submission)

Review form: Reviewer 1 (Lilian Konicar)

Is the manuscript scientifically sound in its present form?

Yes

Are the interpretations and conclusions justified by the results?

Yes

Is the language acceptable?

Yes

Is it clear how to access all supporting data?

Not Applicable

Do you have any ethical concerns with this paper?

No

Have you any concerns about statistical analyses in this paper?

No

Recommendation?

Major revision is needed (please make suggestions in comments)

Comments to the Author(s)

The authors present an interesting approach to investigate the effects of psychopathic characteristics on group cooperative behaviors. The manuscript is not written very concisely and would benefit from an improved academic wording in general. Overall, I think this study is interesting and contributes to the understanding of the effects of individual (psychopathic) personality characteristic on group dynamics, but in an improved version.

ABSTRACT

-) It is not necessary to state the limitations already in the abstract („Despite the small range of participants..“)

-) A revision of the wording would improve the quality of the abstract

INTRODUCTION

The introduction is comprehensible; the red line from exogenous mechanisms to individual characteristics and from cooperation to group dynamics is clear and nice.

-) Why is Ref 11 not included in [8-10]? Could be [8-11], right?

-) The paper would benefit from a comma check (e.g. line 52)

-) I am not sure if “the highest VARIATION” of psychopathy could be found in prisons; the highest scores probably; this first sentence (line 6/ on page 3) could be omitted anyway

-) The whole first paragraph on page 3 (line6-19) does not belong to the introduction and is redundant when reading further the methods part. The wording (including stating in the beginning the research question, then again the hypothesis) does not sound like a sophisticated, academic wording. Please improve.

METHODS

-) As written in the last comment before: please include the whole last paragraph from the introduction part here

-) You wrote “..the first two sessions were pilot sessions..” What were those pilot sessions for? Please explain.

-) Either you capitalize “M” in “Personality measures” like you did it with the “D” in “Experimental Design” or you keep it written in lowercases, but pls standardize it.

-) The paragraphs “Experimental Design” and “Procedure” and intermixed. Please optimize the structure and make it make it more neat and clean.

One paragraph (e.g. “The Prisoners Dilemma”) should describe solely the method, the used experiment – the modified version of the Prisoners Dilemma (until last line 9 on page 4).

Another paragraph (e.g. “Experimental Manipulation: Conditions”) could describe the manipulation – the three groups (starting from line 10); while another paragraph (e.g. “Experimental Procedure”) really describe only the procedural elements.

-) Table 1: please improve the layout

-) The word “Condition” would fit in the whole manuscript more than the word “treatment”

-) To avoid a misleading interpretation of the word “baseline” you could only use “zero density” throughout the whole manuscript

RESULTS

-) There is no citation in the text for figure 1

-) What is the index / score of “Cooperation” in figure 1 and 2 ? How is it calculated?

-) Placing figure 1 +2 next to each other would fit better than figure 2 next to Table 2

-) Please include figure titles
-) The few relevant results/ numbers from table 2 could also be cited in the text and would not need an extra table.
-) Please improve wording (e.g. "INDEED" ...statistically significant" (line 46/ page 6) does not sound very professional academic ("indeed" could be omitted)
-) Table 3: Please improve this table. What does DV, (1), (2) etc. means?
-) I would be interested in social demographics from the sample related to the variables in table 3 (like age, gender, nationality)
-) Please improve the paragraph on page 6 and adopt it to a journal wording
-) Page 7/ Line 49 - the repetition of the focus of the experiment/ analyses could be omitted (line 49 till middle of line 52); also the following paragraph would benefit from rewording
-) Page 7/ Paragraph starting from line 35: this is very interesting, but the results part should only list your results. The comparison, to Rand et al. 2014 and the replication should be part of the discussion and not results.

DISCUSSION

-) The end of game effect is very interesting. A broader interpretation or discussion, why the effect could be observed in the zero density condition, but not in the more psychopathic other conditions would be interesting.
-) You wrote, that your sample had a very restricted psychopathy range compared to the general population. Did you compare statistically or how do you know?
-) As stated in the Methods part, I would be interested in the social demographics, especially of the -as high defined- psychopathic students and the related discussion of it here. If more male students were defined as psychopathic, did you analyze possible psychopathic gender effects? The same question regarding age and nationality would also be interesting.
-) In the discussion I miss a bit a further discussion of the results regarding general implications for groups in different settings like business, academia etc. Who could learn from your results? What does it mean for organizations? For leadership styles, CEOs, successful psychopathy etc.? How do the results fit in the background literature and or regarding minority/majority opinions and changes and the related group behavior and dynamics?

REFERENCES

-) Ref 10: is it 2001 or 2007 ? What about this 7?

Review form: Reviewer 2 (Joshua Plotkin)

Is the manuscript scientifically sound in its present form?

Yes

Are the interpretations and conclusions justified by the results?

Yes

Is the language acceptable?

Yes

Is it clear how to access all supporting data?

Yes

Do you have any ethical concerns with this paper?

No

Have you any concerns about statistical analyses in this paper?

No

Recommendation?

Accept with minor revision (please list in comments)

Comments to the Author(s)

How group composition affects cooperation in fixed networks: can psychopathic traits influence group dynamics?

15th November, 2018

In this submission, the authors experimentally investigate the effects that the so-called psychopathic traits have on the levels of sustained cooperation in a general student population playing a standard Donation Game. "Psychopathic traits" include poor impulse control, lack of guilt, dishonesty and lack of empathy. The authors use an experimental setup that has previously been shown by Rand et al. to support stable cooperation — players arranged on a ring (graph degree $k=2$) with the benefit-to-cost ratio $b/c=6$. The psychopathy score was evaluated using the PPI-R-40 questionnaires, and cooperation levels in three groups with 0%, 20% and 51% of highly psychopathic individuals (the top quartile of the population) was experimentally measured.

The authors demonstrate a significant difference in cooperation levels between groups with 0% and 51% psychopathic individuals (Figure 1), and they find a correlation between cooperation and the fearless-dominance component of players' personality. Interestingly, players trying to maximize pairwise payoffs (their own and their co-players) cooperated more than those maximizing only their own personal payoffs (note that the theoretical $b/c > k$ was derived for the case of imitation based on personal, not pairwise, payoffs). In conclusion, the authors confirm their hypothesis that psychopathic traits affect group dynamics of cooperation, but a substantial density (50%) of individuals with relatively pronounced psychopathic traits was needed to see the effect.

The manuscript presents a significant result, it technically sound and follows the appropriate experimental and data open-access policies. We therefore recommend publication in RS Open Science after a minor revision.

Additional comments and questions:

Using a comparable experimental procedure Rand et al. observed that levels of cooperation do not depend on the round number (their Fig 5), whereas in this report the cooperation frequency declines from 0.7 to 0.55, resembling well-mixed population results of Rand et al. If the end-of-the game effect is to blame, why is it not significant in the other two experimental treatments, and why was it not observed by Rand? How can these effects be avoided in future experiments? How does the distribution of cooperators and defectors change during the course of the experiment, i.e. do clusters similar to Rand's Figure 4 form?

The number of participants per session is much smaller than used by Rand et al. (5-11 vs 24). How was this number chosen prior to the experiments? Did the same individual participate in more than one experimental session? And if so, did they learn to cooperate more or less over time?

It is interesting that fearless dominance, which generally reflects boldness, is associated with lower levels of cooperation within the group. What could potentially explain this effect? Do these individuals cooperate less themselves or is this a secondary effect?

With a small population of 55 to 73 players arranged on a cycle, the relative distribution of the two kinds of players must be important. Would levels of cooperation be lower in with clusters of psychopathic individuals compared to the case where they are evenly distributed along the cycle?

Decision letter (RSOS-181329.R0)

17-Dec-2018

Dear Miss Testori,

The editors assigned to your paper ("How group composition affects cooperation in fixed networks: can psychopathic traits influence group dynamics?") have now received comments from reviewers. We would like you to revise your paper in accordance with the referee and Associate Editor suggestions which can be found below (not including confidential reports to the Editor). Please note this decision does not guarantee eventual acceptance.

Please submit a copy of your revised paper before 09-Jan-2019. Please note that the revision deadline will expire at 00.00am on this date. If we do not hear from you within this time then it will be assumed that the paper has been withdrawn. In exceptional circumstances, extensions may be possible if agreed with the Editorial Office in advance. We do not allow multiple rounds of revision so we urge you to make every effort to fully address all of the comments at this stage. If deemed necessary by the Editors, your manuscript will be sent back to one or more of the original reviewers for assessment. If the original reviewers are not available, we may invite new reviewers.

- Data accessibility

It is a condition of publication that all supporting data are made available either as supplementary information or preferably in a suitable permanent repository. The data

accessibility section should state where the article's supporting data can be accessed. This section should also include details, where possible of where to access other relevant research materials such as statistical tools, protocols, software etc can be accessed. If the data have been deposited in an external repository this section should list the database, accession number and link to the DOI for all data from the article that have been made publicly available. Data sets that have been deposited in an external repository and have a DOI should also be appropriately cited in the manuscript and included in the reference list.

If you wish to submit your supporting data or code to Dryad (<http://datadryad.org/>), or modify your current submission to dryad, please use the following link:
<http://datadryad.org/submit?journalID=RSOS&manu=RSOS-181329>

- **Competing interests**

- **Authors' contributions**

- **Acknowledgements**

- **Funding statement**

on behalf of Dr Joshua Buckholtz (Associate Editor) and Antonia Hamilton (Subject Editor)
openscience@royalsociety.org

Comments to Author:

Reviewers' Comments to Author:

Reviewer: 1

Comments to the Author(s)

The authors present an interesting approach to investigate the effects of psychopathic characteristics on group cooperative behaviors. The manuscript is not written very concisely and would benefit from an improved academic wording in general. Overall, I think this study is interesting and contributes to the understanding of the effects of individual (psychopathic) personality characteristic on group dynamics, but in an improved version.

ABSTRACT

-) It is not necessary to state the limitations already in the abstract („Despite the small range of participants..“)
-) A revision of the wording would improve the quality of the abstract

INTRODUCTION

The introduction is comprehensible; the red line from exogenous mechanisms to individual characteristics and from cooperation to group dynamics is clear and nice.

-) Why is Ref 11 not included in [8-10]? Could be [8-11], right?
-) The paper would benefit from a comma check (e.g. line 52)
-) I am not sure if “the highest VARIATION” of psychopathy could be found in prisons; the highest scores probably; this first sentence (line 6/ on page 3) could be omitted anyway
-) The whole first paragraph on page 3 (line6-19) does not belong to the introduction and is redundant when reading further the methods part. The wording (including stating in the beginning the research question, then again the hypothesis) does not sound like a sophisticated, academic wording. Please improve.

METHODS

-) As written in the last comment before: please include the whole last paragraph from the introduction part here
 -) You wrote “..the first two sessions were pilot sessions..” What were those pilot sessions for? Please explain.
 -) Either you capitalize “M” in “Personality measures” like you did it with the “D” in “Experimental Design” or you keep it written in lowercases, but pls standardize it.
 -) The paragraphs “Experimental Design” and “Procedure” and intermixed. Please optimize the structure and make it more neat and clean.
- One paragraph (e.g. “The Prisoners Dilemma”) should describe solely the method, the used experiment – the modified version of the Prisoners Dilemma (until last line 9 on page 4). Another paragraph (e.g. “Experimental Manipulation: Conditions” could describe the manipulation – the three groups (starting from line 10); while another paragraph (e.g. “Experimental Procedure”) really describe only the procedural elements.
-) Table 1: please improve the layout
 -) The word “Condition” would fit in the whole manuscript more than the word “treatment”
 -) To avoid a misleading interpretation of the word “baseline” you could only use “zero density” throughout the whole manuscript

RESULTS

-) There is no citation in the text for figure 1
-) What is the index / score of “Cooperation” in figure 1 and 2 ? How is it calculated?
-) Placing figure 1 +2 next to each other would fit better than figure 2 next to Table 2
-) Please include figure titles

-) The few relevant results/ numbers from table 2 could also be cited in the text and would not need an extra table.
-) Please improve wording (e.g. "INDEED" ...statistically significant" (line 46/ page 6) does not sound very professional academic ("indeed" could be omitted)
-) Table 3: Please improve this table. What does DV, (1), (2) etc. means?
-) I would be interested in social demographics from the sample related to the variables in table 3 (like age, gender, nationality)
-) Please improve the paragraph on page 6 and adopt it to a journal wording
-) Page 7/ Line 49 – the repetition of the focus of the experiment/ analyses could be omitted (line 49 till middle of line 52); also the following paragraph would benefit from rewording
-) Page 7/ Paragraph starting from line 35: this is very interesting, but the results part should only list your results. The comparison, to Rand et al. 2014 and the replication should be part of the discussion and not results.

DISCUSSION

-) The end of game effect is very interesting. A broader interpretation or discussion, why the effect could be observed in the zero density condition, but not in the more psychopathic other conditions would be interesting.
-) You wrote, that your sample had a very restricted psychopathy range compared to the general population. Did you compare statistically or how do you know?
-) As stated in the Methods part, I would be interested in the social demographics, especially of the -as high defined- psychopathic students and the related discussion of it here. If more male students were defined as psychopathic, did you analyze possible psychopathic gender effects? The same question regarding age and nationality would also be interesting.
-) In the discussion I miss a bit a further discussion of the results regarding general implications for groups in different settings like business, academia etc. Who could learn from your results? What does it mean for organizations? For leadership styles, CEOs, successful psychopathy etc.? How do the results fit in the background literature and or regarding minority/majority opinions and changes and the related group behavior and dynamics?

REFERENCES

-) Ref 10: is it 2001 or 2007 ? What about this 7?

Reviewer: 2

Comments to the Author(s)

How group composition affects cooperation in fixed networks: can psychopathic traits influence group dynamics?

15th November, 2018

In this submission, the authors experimentally investigate the effects that the so-called psychopathic traits have on the levels of sustained cooperation in a general student population playing a standard Donation Game. "Psychopathic traits" include poor impulse control, lack of guilt, dishonesty and lack of empathy. The authors use an experimental setup that has previously been shown by Rand et al. to support stable cooperation – players arranged on a ring (graph degree $k=2$) with the benefit-to-cost ratio $b/c=6$. The psychopathy score was evaluated using the PPI-R-40 questionnaires, and cooperation levels in three groups with 0%, 20% and 51% of highly psychopathic individuals (the top quartile of the population) was experimentally measured.

The authors demonstrate a significant difference in cooperation levels between groups with 0% and 51% psychopathic individuals (Figure 1), and they find a correlation between cooperation

and the fearless-dominance component of players' personality. Interestingly, players trying to maximize pairwise payoffs (their own and their co-players) cooperated more than those maximizing only their own personal payoffs (note that the theoretical $b/c > k$ was derived for the case of imitation based on personal, not pairwise, payoffs). In conclusion, the authors confirm their hypothesis that psychopathic traits affect group dynamics of cooperation, but a substantial density (50%) of individuals with relatively pronounced psychopathic traits was needed to see the effect.

The manuscript presents a significant result, it is technically sound and follows the appropriate experimental and data open-access policies. We therefore recommend publication in RS Open Science after a minor revision.

Additional comments and questions:

Using a comparable experimental procedure Rand et al. observed that levels of cooperation do not depend on the round number (their Fig 5), whereas in this report the cooperation frequency declines from 0.7 to 0.55, resembling well-mixed population results of Rand et al. If the end-of-the game effect is to blame, why is it not significant in the other two experimental treatments, and why was it not observed by Rand? How can these effects be avoided in future experiments? How does the distribution of cooperators and defectors change during the course of the experiment, i.e. do clusters similar to Rand's Figure 4 form?

The number of participants per session is much smaller than used by Rand et al. (5-11 vs 24). How was this number chosen prior to the experiments? Did the same individual participate in more than one experimental session? And if so, did they learn to cooperate more or less over time?

It is interesting that fearless dominance, which generally reflects boldness, is associated with lower levels of cooperation within the group. What could potentially explain this effect? Do these individuals cooperate less themselves or is this a secondary effect?

With a small population of 55 to 73 players arranged on a cycle, the relative distribution of the two kinds of players must be important. Would levels of cooperation be lower in with clusters of psychopathic individuals compared to the case where they are evenly distributed along the cycle?

Author's Response to Decision Letter for (RSOS-181329.R0)

See Appendix A.

Decision letter (RSOS-181329.R1)

29-Jan-2019

Dear Miss Testori,

I am pleased to inform you that your manuscript entitled "How group composition affects cooperation in fixed networks: can psychopathic traits influence group dynamics?" is now accepted for publication in Royal Society Open Science.

on behalf of Dr Joshua Buckholtz (Associate Editor) and Antonia Hamilton (Subject Editor)
openscience@royalsociety.org

Follow Royal Society Publishing on Twitter: [@RSocPublishing](https://twitter.com/RSocPublishing)
Follow Royal Society Publishing on Facebook:
<https://www.facebook.com/RoyalSocietyPublishing.FanPage/>
Read Royal Society Publishing's blog: <https://blogs.royalsociety.org/publishing/>

Appendix A

We thank the reviewers for their careful reading of our manuscript and suggestions for improvements. We have addressed the comments of the reviewers and editor as detailed below.

Reviewer: 1

ABSTRACT

-) It is not necessary to state the limitations already in the abstract („Despite the small range of participants..“)
-) A revision of the wording would improve the quality of the abstract

We have removed the phrase “Despite the small range of participants’ psychopathy scores” from the abstract following your suggestion.

INTRODUCTION

The introduction is comprehensible; the red line from exogenous mechanisms to individual characteristics and from cooperation to group dynamics is clear and nice.

-) Why is Ref 11 not included in [8-10]? Could be [8-11], right?
-) The paper would benefit from a comma check (e.g. line 52)
-) I am not sure if “the highest VARIATION” of psychopathy could be found in prisons; the highest scores probably; this first sentence (line 6/ on page 3) could be omitted anyway
-) The whole first paragraph on page 3 (line6-19) does not belong to the introduction and is redundant when reading further the methods part. The wording (including stating in the beginning the research question, then again the hypothesis) does not sound like a sophisticated, academic wording. Please improve.

We have followed your suggestions and we modified the references and the wording throughout the entire section. Please find the changes highlighted in the manuscript.

METHODS

-) As written in the last comment before: please include the whole last paragraph from the introduction part here

We have eliminated the last paragraph from the introduction and we did not include it in the methods section as redundant.

-) You wrote “..the first two sessions were pilot sessions..” What were those pilot sessions for? Please explain.

We now explain that the first two sessions were run as pilot sessions to check the functionality of the experiment. (First paragraph of the Methods section, page 3).

-) Either you capitalize “M” in “Personality measures” like you did it with the “D” in “Experimental Design” or you keep it written in lowercases, but pls standardize it.

We have now made the capitalisation of the section titles consistent.

-) The paragraphs “Experimental Design” and “Procedure” are intermixed. Please optimize the structure and make it more neat and clean. One paragraph (e.g. “The Prisoners Dilemma”) should describe solely the method, the used experiment – the modified version of the Prisoners Dilemma (until last line 9 on page 4). Another paragraph (e.g. “Experimental Manipulation: Conditions”) could describe the manipulation – the three groups (starting from line 10); while another paragraph (e.g. “Experimental Procedure”) really describe only the procedural elements.

We have now reorganised these sections to make a clearer distinction of the contents.

We split Experimental Design section into 2 sections (Experimental Design & Experimental Manipulation).

-) Table 1: please improve the layout

We have now centred the Table and we checked the layout of all the Tables in the manuscript.

-) The word “Condition” would fit in the whole manuscript more than the word “treatment”

We now use the term Condition throughout the whole manuscript.

-) To avoid a misleading interpretation of the word “baseline” you could only use “zero density” throughout the whole manuscript

To avoid confusion, we now use the term zero density throughout the entire manuscript.

RESULTS

-) There is no citation in the text for figure 1

Figure 1 is cited in the second sentence of the Results section and before Table 3.

-) What is the index / score of “Cooperation” in figure 1 and 2? How is it calculated?

The cooperation variable presented in Figures 1 and 2 is the same variable used in the regression analysis and described in the first sentence of the Results section.

We have now made it clearer by stating it in the Figures captions.

-) Placing figure 1 +2 next to each other would fit better than figure 2 next to Table 2

We have now re-organised the order of the Figures.

-) Please include figure titles

We have now included titles for both Figures.

-) The few relevant results/ numbers from table 2 could also be cited in the text and would not need an extra table.

Thank you for your suggestion. We would like to keep the table to emphasize the correlations amongst the dependent variable and psychopathic traits.

-) Please improve wording (e.g. “INDEED” ...statistically significant” (line 46/ page 6) does not sound very professional academic (“indeed” could be omitted)

-) Table 3: Please improve this table. What does DV, (1), (2) etc. means?

DV stands for dependent variable. We have omitted it in the table and we specify the dependent variable in the Table caption. We also explicitly stated model (1), (2) etc.

-) I would be interested in social demographics from the sample related to the variables in table 3 (like age, gender, nationality)

We have now included a table describing the sample demographics (Table 1).

-) Please improve the paragraph on page 6 and adopt it to a journal wording

-) Page 7/ Line 49 – the repetition of the focus of the experiment/ analyses could be omitted (line 49 till middle of line 52); also the following paragraph would benefit from rewording

We decided to keep the repetition of the focus of the experiment to give the reader a better guidance through the results.

-) Page 7/ Paragraph starting from line 35: this is very interesting, but the results part should only list your results. The comparison, to Rand et al. 2014 and the replication should be part of the discussion and not results.

We have now moved the comments related the comparison between our and Rand et al. (2014) results to the discussion (fourth paragraph).

DISCUSSION

-) The end of game effect is very interesting. A broader interpretation or discussion, why the effect could be observed in the zero density condition, but not in the more psychopathic other conditions would be interesting.

We have now broadened the discussion of the end of game effect (fourth paragraph).

-) You wrote, that your sample had a very restricted psychopathy range compared to the general population. Did you compare statistically or how do you know?

We have now included the comparison of our sample with two other studies. We included a short paragraph describing this in the Personality Measures sub-section and we included the relevant statistics in Table 1.

-) As stated in the Methods part, I would be interested in the social demographics, especially of the -as high defined- psychopathic students and the related discussion of it here. If more male students were defined as psychopathic, did you analyze possible psychopathic gender effects? The same question regarding age and nationality would also be interesting.

We have now included the correlation matrix between psychopathic measures and demographic information in the Results section (Table 3). Also, demographic variables were already included in the regressions to check for possible divergences in the groups, and we commented those differences in the Results section (only nationality had an effect on cooperation). Moreover, we now added a 5th model in the regression showing that belonging to a group with a higher concentration of high psychopathic individuals (i.e. high density condition) is the driving force towards more defective behaviours, regardless of individuals' psychopathic attitudes (Table 4).

Also, the correlation showed in Table 3 is consistent with the literature: male subjects are known to exhibit higher levels of psychopathic traits compared to female ones.

-) In the discussion I miss a bit a further discussion of the results regarding general

implications for groups in different settings like business, academia etc. Who could learn from your results? What does it mean for organizations? For leadership styles, CEOs, successful psychopathy etc.? How do the results fit in the background literature and or regarding minority/majority opinions and changes and the related group behavior and dynamics?

We have now included a section in the second paragraph of the Discussion addressing these questions: "This has relevant implications for group settings, e.g. team work in companies or educational environments. On an individual level, psychopathy has been found to be related to counterproductive work behaviour and negative impact on employees. Our results therefore align with negative effects of psychopathic personality traits on individuals in the work context, but extend those findings to less cooperative behaviour in team settings. This could have implications for building and managing teams, especially when cooperative behaviour is crucial for successful team work."

REFERENCES

-) Ref 10: is it 2001 or 2007 ? What about this 7?

We checked the citation and it is correct. The title of the article begins with 7. and the year is 2001.

Reviewer: 2

Using a comparable experimental procedure Rand et al. observed that levels of cooperation do not depend on the round number (their Fig 5), whereas in this report the cooperation frequency declines from 0.7 to 0.55, resembling well-mixed population results of Rand et al. If the end-of-the game effect is to blame, why is it not significant in the other two experimental treatments, and why was it not observed by Rand?

Thank you very much for the interesting question.

One hypothesis could be that the number of individuals in the session could be determinant to observe an end-of-the-game effect. In our experiment groups were composed of maximum 11 players while in the static network treatment of Rand et al. there are roughly 24 players per session.

With respect to the differences in our treatments:

- 1) The pattern for the low and zero density treatments is very similar (Figure 2) and the absence of a statistically significant level in the low treatment could be due to the smaller sample size (55 VS 62);
- 2) Groups in the high treatment adopted a strongly defective behaviour and it is possible that they were not affected by the end-of-game effect because of the already low level of cooperation.

We have now added a paragraph in the Discussion about the end-of-game effect (fourth paragraph).

How can these effects be avoided in future experiments?

A possible solution for this problem is to avoid showing the round number to players or to end the game after a random number of rounds. Both techniques have been largely used in previous experiments. Although we did not show the round number to participants, they knew that the game was formed of 50 rounds, so they could have counted or simply guessed when the game was about to end.

How does the distribution of cooperators and defectors change during the course of the experiment, i.e. do clusters similar to Rand's Figure 4 form?

We did not analyse the distribution of cooperators and defectors in the network.

The aim of our study is to see how differently composed groups behave over time.

By looking at the evolution of individuals' strategy over time, the main focus of the manuscript would shift. Nevertheless, this is of great interest and it is something we would like to address in future research, as mentioned in the discussion (sixth paragraph).

The number of participants per session is much smaller than used by Rand et al. (5-11 vs 24). How was this number chosen prior to the experiments?

Rand and colleagues ran two different experiments – a lab experiment and a computer experiment. In the lab experiment (which is more similar to the one we replicated with respect to the layout) the number of participants per session was roughly the same (8.4 subjects per session on average).

As we conducted the experiment in a lab, we could not replicate sessions with 24 participants, hence, we decided to adopt a similar number to that in the lab experiment that Rand et al. ran.

Did the same individual participate in more than one experimental session? And if so, did they learn to cooperate more or less over time?

Each participant took part in a single experimental session, never in more sessions.

We have made it explicit also in the manuscript now.

It is interesting that fearless dominance, which generally reflects boldness, is associated with lower levels of cooperation within the group. What could potentially

explain this effect? Do these individuals cooperate less themselves or is this a secondary effect?

Thank you for the very interesting comment. It could be explained by the fact that individuals who show lower cooperative behaviour are less concerned about upsetting their fellow players. Indeed, those players are more fearless, more stress tolerant and more socially dominant (these are the characteristics of fearless dominance), which makes them caring less about the others.

With a small population of 55 to 73 players arranged on a cycle, the relative distribution of the two kinds of players must be important. Would levels of cooperation be lower in with clusters of psychopathic individuals compared to the case where they are evenly distributed along the cycle?

Thank you for your comment. In our experiment, the participants were arranged on the ring so that there were no clusters of high psychopathic individuals. They were always evenly distributed along the cycle. For this reason, we cannot compare the two situations you mentioned, although we think it would be an interesting point for further studies. We have now specified the arrangement of players on the cycle better in the Experimental Manipulation sub-section of the Methods section.